# Genomic Insights into Tumorigenesis in Newly Diagnosed Multiple Myeloma

**DOI:** 10.3390/diagnostics15172130

**Published:** 2025-08-23

**Authors:** Marina Kyriakou, Costas Papaloukas

**Affiliations:** Department of Biological Applications and Technology, University of Ioannina, GR45110 Ioannina, Greece; mailto:kyriakoumarin@gmail.com

**Keywords:** plasma cell dyscrasias, multiple myeloma, single-cell RNA-sequencing, bioinformatics, germline mutations, somatic mutations, carcinogenesis, disease progression

## Abstract

**Background**: Multiple Myeloma (MM) is a malignant plasma cell dyscrasia that progresses through the consecutive asymptomatic, often undiagnosed, precancerous stages of Monoclonal Gammopathy of Undetermined Significance (MGUS) and Asymptomatic Multiple Myeloma (SMM). MM is characterized by low survival rates, severe complications and drug resistance; therefore, understanding the molecular mechanisms of progression is crucial. This study aims to detect genetic mutations, both germline and somatic, that contribute to disease progression and drive tumorigenesis at the final stage of MM, using samples from patients presenting MGUS or SMM, and newly diagnosed MM patients. **Methods**: Mutations were identified through a fully computational pipeline, implemented in a Linux and RStudio environment, applied to each patient sequence, obtained through single-cell RNA-sequencing (scRNA-seq), separately. Structural and functional mutation types were identified by stage, along with the affected genes. The analysis included quality control, removal of the Unique Molecular Identifiers (UMIs), trimming, genome mapping and result visualization. **Results**: The findings revealed frequent germline and somatic mutations, with distinct structural and functional patterns across disease stages. Mutations in key genes were identified, pointing to molecules that may play a central role in carcinogenesis and disease progression. Notable examples include the *HLA-A*, *HLA-B* and *HLA-C* genes, as well as the *KIF*, *EP400* and *KDM* gene families, with the first four already confirmed. Comparative analysis between the stages highlighted molecular transition events from one stage to another. Emphasis was given to novel genes discovered in newly diagnosed MM patients, that might contribute to the tumorigenesis that takes place. **Conclusions**: This study contributes to the understanding of the genetic basis of plasma cell dyscrasias and the transition events between the stages, offering insights that could aid in early detection and diagnosis, guide the development of personalized therapeutic strategies, and improve the understanding of mechanisms responsible for resistance to existing therapies.

## 1. Introduction

Plasma cell dyscrasias are pathological hematologic conditions comprising a heterogeneous group of diseases characterized by clonal proliferation of bone marrow plasma cells and production of a monoclonal immunoglobulin, known as the M-protein, detected as a paraprotein present in serum and or urine [1]. These disorders encompass a broad variety of conditions, ranging from asymptomatic precancerous stages, such as Monoclonal Gammopathy of Undetermined Significance (MGUS) and Asymptomatic/Smoldering Multiple Myeloma (SMM), to malignant diseases, including Multiple Myeloma (MM) and plasma cell leukemia [2]. MGUS is found in approximately 5% of the population over the age of 50, and although it may remain stable for an extended period of time, it has the potential to progress to MM at an annual rate of 1% [3]. SMM represents an intermediate asymptomatic condition in the progression from MGUS to MM, with all MM cases arising from SMM. This stage differs from MGUS by its higher risk of progression to MM within the first five years post diagnosis [3]. As the disease advances, each heterogenous stage is characterized not only by increasing amounts of clonal bone marrow plasma cells and higher levels of M-protein, but also by differences in the type of M-protein. The stages also display variability in genetic changes, including cytogenetic abnormalities, such as translocations and trisomies, which may contribute to disease progression, among other things [4]. However, MM is distinguished from its precursor stages by severe bone destruction, which appears as osteolytic lesions, compression fractures, and skeletal weakening [2]. It is worth mentioning that both pre-malignant conditions are difficult to diagnose [5].

Although the pathogenesis of MM is complex and not yet fully elucidated, it is thought to result from a combination of alterations in the genetic material and changes in the bone marrow microenvironment [2]. Despite significant novel therapeutic advancements that can induce remission, MM remains incurable [6] with most patients experiencing relapse and eventually developing drug resistance [1]. Genetic mutations, both germline and somatic, are key drivers of the development and progression of malignant conditions, including MM and its precursor stages [6]. Germline mutations originate in reproductive cells, during gametogenesis [7], are inherited, and can therefore increase an individual’s predisposition to malignancy, without directly causing it. On the other hand, somatic mutations arise in non-reproductive cells during the individual’s life, due to either random errors or environmental factors [7] and drive malignant transformation and disease advancement [8]. These mutations can be classified structurally, as single nucleotide polymorphisms (SNPs) and insertions/deletions (INDELs) [9], and functionally, which include but are not limited to, missense, frameshift, start-loss or stop-loss and splice variants [10], both categories may lead to the development of diseases and the alterations of functions of certain proteins that impair essential cellular processes [9].

Next-generation sequencing (NGS) technology has revolutionized the detection and characterization of such mutations, enabling high-throughput, accurate analysis of large amounts of genetic information [11] and providing deep insights into the structure and complexity of the genome [9]. Among NGS applications, one of the most used is RNA-sequencing (RNA-seq), as it has proven particularly useful in analyzing gene expression [12] and detecting low-frequency mutations, which may not be identified using the conventional method of DNA-sequencing [12]. However, single-cell RNA-sequencing (scRNA-seq), as an emerging sequencing technology [13], has become a powerful, yet still developing tool that allows for the comparative analysis of individual cell transcriptomes, revealing transcriptional differences and similarities within a specific cell population, under both physiological and pathological conditions [14]. This method stands out due to its ability to detect rare cell subpopulations, such as tumor cancer cells [14], which may be masked by standard RNA-seq methods [15]. Single-cell studies have further revealed the heterogeneity present among different subpopulations of cancer cells within the same tumor, as well as the long-term evolution of tumors in MM [16]. Additionally, both approaches support the use of Unique Molecular Identifiers (UMIs), which allow for distinguishing original molecules from duplicates that may arise during Polymerase Chain Reaction (PCR) amplification, thereby enhancing the quantitative accuracy of both RNA-seq and scRNA-seq, especially in samples with low RNA content [17]. UMIs integration reduces false positive results due to duplicate reads and enhances the accuracy of variant detection [18].

Given the challenges in early diagnosis and the limited preventability of MM, estimated at just 14% of cases [19], the identification of molecular mechanisms driving progression from MGUS and SMM to MM remains critically important [5]. To further highlight the clinical significance of MM, in 2022, Western Europe accounted for 9% of all MM cases, with Northern Europe showing some of the highest mortality rates, namely 1.8 per 100,000, as per GLOBACAN 2022. Globally, if current rates remain unchanged, the incidence and mortality of MM are projected to increase by 71% and 79%, respectively, by 2045 [20]. This study aims to identify and compare germline and somatic genetic variants associated with MM, by analyzing scRNA-seq data from patients across the plasma cell dyscrasias spectrum, namely MGUS, SMM and newly diagnosed MM. The analysis focused on detecting and comparing SNPs and INDELs, aiming to elucidate early molecular events potentially contributing to MM pathogenesis. To achieve this, a completely computational pipeline was implemented, integrating quality control, sequence trimming and removal of UMIs, genome mapping, germline and somatic variant detection and annotation, as well as comparative evaluation of the results between each disease, respectively, and across the transitions between the stages, providing invaluable insights into the molecular changes underlying progression and carcinogenesis in the MM stage. This bioinformatics approach revealed distinct mutational patterns and discovered several novel candidate genes, to our best of knowledge, not previously implicated in MM, suggesting their role in carcinogenesis and disease evolution, as well as in future therapeutic targets directing attention to the early phases of MM.

These findings provide valuable information on the plasma cell dyscrasias, specifically the malignant disease of MM, and may play a central role in the development of predictive markers, both at the precancerous stages and at the tumorigenesis one. They may also aid in creating personalized treatment strategies, as well as in early prognosis and identification of the mechanisms responsible for drug resistance.

## 2. Materials and Methods

### 2.1. Data Acquisition

To identify mutations potentially responsible for the progression of Multiple Myeloma (MM) and its precursor stages, a comparative analysis was performed between individuals at each premalignant stage, namely Monoclonal Gammopathy of Undetermined Significance (MGUS) and Smoldering Multiple Myeloma (SMM), newly diagnosed patients with Multiple Myeloma (MM) and healthy individuals without diagnosis of any of the diseases mentioned in this study. The latter served as control samples, providing a reference point for identifying mutations present exclusively in pathological samples.

All datasets analyzed in this study were obtained from the publicly available Gene Expression Omnibus (GEO) database, specifically from the Series GSE271107, which includes all sequenced data used herein. The datasets included four healthy controls (GSM8369864, GSM8369865, GSM8369866, GSM8369867), four patients with MGUS diagnosis (GSM8369868, GSM8369869, GSM8369870, GSM8369871), four SMM patients (GSM8369874, GSM8369875, GSM8369876, GSM8369877) and four newly diagnosed MM patients (GSM8369878, GSM8369879, GSM8369880, GSM8369881).

All samples had been previously sequenced using the Illumina Novaseq 6000 (GPL24676) platform, a high-throughput scRNA-sequencing system with low error rates and high yield [21]. This platform follows the standard Illumina workflow [22], which includes reverse transcription of RNA into complementary DNA (cDNA), library preparation, cDNA amplification via Polymerase Chain Reaction (PCR), and sequencing by synthesis. For this study, library preparations were carried out using the Chromium Controller 10×Genomics (Pleasanton, CA, USA), which integrates Unique Molecular Identifiers (UMIs) following the cell-specific barcodes during library construction. Each sample was also registered in the Sequence Read Archive (SRA) database under a unique Experiment Accession, SRX, with four independent sequencing runs, namely Run Accessions SRR, per SRX. The data used in this study are publicly available under the Study Accession PRJNA1129864. For each sample, the SRR with the highest number of bases was selected, as higher coverage is indicative of better data quality. Because the size of each SRR file exceeds the 5 GB limit imposed by the SRA database, the FASTQ files were retrieved using the SRA Toolkit on Ubuntu Linux version 22.04.5 LTS (Jammy Jellyfish).

The rest of the preprocessing was also carried out on Ubuntu Linux, selected for its widespread utilization in bioinformatics, ease of use, as well as compatibility with popular bioinformatics tools [23]. Notably, when any errors or warnings arose during the next steps that could not be resolved conventionally, generative artificial intelligence (GenAI) was employed to assist in troubleshooting.

Figure 1 illustrates the computational pipeline implemented in this study for the preprocessing of raw sequence data, as well as for the variant calling processes. This workflow generated the necessary data for the analysis of mutations in cancer patients and their precursor stages.

### 2.2. File Preparation

Each SRR download contains four different FASTQ files. The two smaller files, called Index 1 (I1) and Index 2 (I2), store only index sequences and are used exclusively during demultiplexing to separate sample reads when converting raw BCL files into usable FASTQ files. These were not relevant to the computational pipeline of this study and thus were excluded. The remaining two files are named Read 1 (R1) and Read 2 (R2) and were utilized in this analysis. Read 1 is 28 base pairs (bp) long and contains the cell barcodes, that take up 16 bp, followed by 12 bp Unique Molecular Identifiers (UMIs) [24]. The R2 is the largest file, as it contains exclusively the cDNA sequence that consists of 90 bp, along with the UMI information from R1.

For this analysis, only R1 and R2 were used, with particular emphasis on R2, as it contains the cDNA sequence required for the study objectives. Although FASTQ file compression does not affect the process or the analysis results, the files were compressed solely to save storage space, since each file exceeded 60 GB.

A quality control assessment was performed on the R2 files of each sample, using the FastQC tool version 0.12.1 [25], and the results were summarized with MultiQC version 1.28. These reports were used only to compare the quality of raw versus trimmed sequences and were not directly included in subsequent analysis.

### 2.3. Unique Molecular Identifiers Tagging and Processing

The short UMI sequences present in R1 were removed from the R2 prior to further pre-processing. The identification and removal of these sequences from R2 was performed using the UMI-tools repository version 1.1.6 [26], and specifically the extract command. When using the whitelist command to enable further filtering and removal of barcodes, the new FASTQ files produced only contained a few kilobytes of information, indicating excessive sequence loss. To preserve data integrity, only the extract command was retained for downstream analysis. This command detects the UMIs in R1, removes them from the R2 sequence, and incorporates them into the read name in the corresponding FASTQ header. The resulting R2 file contains the cDNA sequence, while UMI information is recorded in the header; however, R1 was no longer required for subsequent analysis steps, as it was used exclusively for R2 processing [26].

### 2.4. Read Trimming

The new R2 sequence, with UMI information already removed, was further processed to improve its quality using the tool TrimGalore! version 0.6.1028, with default parameters. This Perl-based wrapper integrates Cutadapt for trimming and FastQC for quality assessment, leveraging the quality indicators of the latter for targeted and automated trimming without the need for additional parameters.

To evaluate whether an alternative trimming approach could yield better results, Cutadapt version 5.0 was also tested separately, targeting only overrepresented sequences. This approach was applied both to the R2 file already trimmed with TrimGalore! and to the R2 file immediately after UMI-tools processing. After trimming, the quality of each R2 file was reassessed using the same two tools mentioned above. Based on this comparison, TrimGalore! alone was selected for the final preprocessing workflow, out of the three tested strategies.

### 2.5. Mapping to the Human Reference Genome

After trimming, the R2 reads were mapped to the GRCh38 reference genome, also known as hg38, as specified by the GEO dataset characteristics. The HISAT2 tool, version 2.2.1, was used for this step, as it provides fast and accurate alignment of RNA-seq and scRNA-seq data [27]. Each output file, initially in SAM format, was then converted to its binary form, BAM, which contains the same information in compressed format to optimize downstream analysis performance.

Subsequently, each BAM file was sorted, and an index was generated for each file; both were carried out using the samtools software version 1.19.2, to ensure compatibility with subsequent analysis steps [28].

The quality of the mapped reads in the sorted BAM files was assessed using Qualimap tool version 2.3 [29]. The resulting metrics included alignment percentages, duplication rates and GC content, providing insight into the data quality and the effectiveness of the alignment process with respect to the reference genome [25].

Typically, a deduplication step follows mapping; however, initial attempts to deduplicate BAM files, to identify and remove potential duplicate reads while taking UMIs into account, using UMI-tools’ command dedup [26], resulted in empty outputs, likely due to the nature of the data. Therefore, this step was omitted to avoid loss of biologically meaningful repetitive regions, which cannot be easily distinguished from technical duplicates in RNA-sequencing, and specifically single-cell RNA-sequencing data [30].

In order to commence the variant detection process, it was necessary to add read group tags to each BAM file, defining the identity of each patient and control sample, to enable proper variant detection [31]. This step was completed using the GATK tools version 4.2.6.1 [31], followed by indexing the updated BAM files [28,31,32].

These pre-processing steps ensured that the data were of sufficient quality and well-prepared for mutation detection stage.

### 2.6. Variant Calling

The germline variant detection, namely the mutations of inherited predisposition, was performed using the tool HaplotypeCaller from GATK, since it is widely regarded as the most accurate for detecting single-nucleotide polymorphisms (SNPs) and small insertions and deletions (INDELs) of germline origin [32]. In conjunction with the reference genome, to identify regions of the genome with variation in the samples relative to the reference [8], a confidence threshold of 30.0 was applied during execution. This ensured that only variants with an estimated error probability below 0.1% were recorded, guaranteeing that the detected variants met a minimum confidence level for validity, being true positives and biologically present [33].

Somatic variant detection was subsequently performed on the same files using Mutect2 tool, also from GATK. These mutations, acquired during the patient’s lifetime, are more frequent in individuals diagnosed with cancer, making this tool suitable for analyzing the samples used in this study [32,34]. Like HaplotypeCaller, Mutect2 also uses the reference genome GRCh38 but does not require specifying a confidence threshold.

### 2.7. Variant Filtering

The previous step produced VCF files for each sample, containing the detected genomic variants, either germline or somatic, identified by HaplotypeCaller or Mutect2, respectively. Each recorded variant, in the VCF files, includes information such as chromosome, exact genomic position, nucleotide change, and confidence score. Filtering was then performed to remove low-confidence variants, optimizing the results.

A common filter in such analyses excludes variants with a Phred Quality Score (QUAL) below 30, corresponding to an error probability of 0.1% and an accuracy of 99.9%. The QUAL, computed by GATK, is inversely related to the likelihood of error [33].

For the VCF files generated by HaplotypeCaller, the GATK VariantFiltration command was applied, marking variants with QUAL < 30.0, as low quality in the FILTER field of the VCF file, without removing them at this stage. This approach, as opposed to earlier threshold-based filtering, allowed flagged variants to be excluded in later stages of analysis [35].

In contrast, the VCF files generated by Mutect2 were filtered using the specialized GATK command FilterMutectCalls, which evaluates each variant based on predefined automated quality criteria.

### 2.8. Variant Selection

Next, SNPs and INDELs were extracted separately from each sample to distinguish between these two major types of genetic variants, using bcftools version 1.19 [36].

The PASS filter was applied to retain only variants that met the previously defined quality criteria. It is important to analyze SNPs and INDELs separately, as they can lead to different types of genetic alterations.

### 2.9. Comparison of Patients and Controls Variants

Comparisons of the detected variants from each patient to each control sample were performed to identify unique variants potentially associated with MM or its precancerous stages. This comparison was performed using the isec command of bcftools [36].

A total of 64 comparisons were performed for each plasma cell disorder analyzed in this study, including 16 for SNPs from the filtered VCFs of HaplotypeCaller, 16 for SNPs from Mutect2, and 32 for INDELs from the VCFs of both tools. For each comparison, four VCF files were generated, namely 0000.vcf contained variants unique to the patient sample, 0001.vcf contained variants unique to the control sample, 0002.vcf contained the shared variants, and 0003.vcf contained the union of all variants.

### 2.10. Annotation of Detected Mutations

After comparing patients and controls, the patient-unique variants from each 0000.vcf file were annotated to determine their functional and biological significance. This step identified the gene in which the variant was located, the potentially affected protein, and the predicted functional impact [37]. Cataloging such pathogenic variants in the human genome is crucial, as it enhances our understanding of disease progression and can aid in detecting the relevant mutations for the study objectives [38].

Annotation in this study was performed using the SnpEff tool, version 5.2f (build on 7 February 2025) [39], which supports both SNPs and INDELs. The tool classifies variants according to their function and assigns predicted impact categories as HIGH, MODERATE, LOW, or MODIFIER [40]. These categories were then used to prioritize biologically meaningful variants in subsequent filtering and analysis.

### 2.11. Organization and Visualization

Upon completion of the annotation of the detected variants, the resulting VCF files were converted to CSV format to facilitate further data analysis and processing, using the *awk* command. During this conversion, it was observed that in some cases the ALT field contained multi-base insertions or multiple alternative alleles, causing column misalignment. To address this, the CSV files were reformatted to ensure that each variant corresponded to a single row, maintaining consistency of values across columns. For each patient–control comparison, the relevant CSVs containing the annotated SNPs and INDELs from both HaplotypeCaller and Mutect2 tools were then merged, resulting in a single consolidated CSV file per comparison that included all annotated variants.

Variants were then filtered by predicted impact, retaining only those classified as HIGH or MODERATE impact, therefore focusing the analysis on the most biologically significant variants.

To identify meaningful patterns, additional classification was performed to highlight mutations consistently present across all comparisons within a stage, as well as those uniquely emerging at each disease transition, namely from MGUS to SMM and from SMM to MM. Additionally, mutations specific to the final MM stage, detected in at least 12 patient–control comparisons, were prioritized for visualization.

To assess whether specific substitution types were statistically enriched in each stage, Fisher’s exact tests were performed for each substitution-stage combination. This test evaluates whether the frequency of a given substitution is significantly associated with a specific disease stage when compared to its distribution in the other stages. Fisher’s statistical test was selected due to its suitability for categorical data across all sample sizes [41], as well as its tendency to yield more conservative probability values (*p*-values), thereby reducing the risk of false positives [42]. Substitutions with a *p*-value of less than 0.05 were considered significantly enriched.

All further analyses and visualizations were carried out in R programming language (version 4.4.3) within the RStudio environment (version 2025.5.0.496). These tools were selected for their ability to efficiently handle and analyze large-scale data using specialized libraries and packages for statistical and bioinformatic analyses [43]. Summary plots included barplots of functional and structural mutation types, SNP substitution patterns, and OncoPrint-style heatmaps [44].

## 3. Results

### 3.1. Quality Control of Raw and Trimmed Reads

The initial quality assessment of the raw Read 2 (R2) reads indicated that none of the samples were flagged as low quality. However, all samples failed the duplication level metric and displayed warnings for the presence of overrepresented small sequences. In addition, most samples showed warnings for per-base sequence content, and only a few failed the Guanine-Cytosine (GC) content metric.

During the trimming process, multiple strategies were evaluated to optimize the quality of the R2 sequences. Among all methods applied, TrimGalore! alone, consistently yielded the highest overall quality compared to the alternative strategies, as reflected in the updated FastQC reports.

Consequently, TrimGalore! was retained as the final trimming method utilized in this pipeline. Nevertheless, all samples failed both the duplication levels and the per-base sequence content metric, while warnings were reported for the presence of overrepresented sequences and for sequence length distribution. In some samples, the GC content metric also remained flagged as failed.

Although the post-trimming FastQC reports initially showed more warnings compared to the raw data, this was expected and does not indicate an actual decrease in quality, as noted in the official FastQC documentation [45]. Rather, these changes reflect the removal of adapters and overrepresented small sequences, which influence certain quality metrics.

### 3.2. BAM Quality Assessment and Alignment Rates

Qualimap analysis of the BAM files confirmed that all samples achieved acceptable alignment rates, duplication levels and average coverage [29], supporting the reliability and suitability of the data for variant detection.

The alignment of the processed reads to the reference genome GRCh38 resulted in overall alignment rates ranging between 83% and 94%, which fall within the expected range for high-quality single-cell RNA-sequencing (scRNA-seq data), namely 70–90% [25]. The Monoclonal Gammopathy of Undetermined Significance (MGUS) exhibited the highest mean overall alignment rates, while Smoldering Multiple Myeloma (SMM) and Multiple Myeloma (MM) samples showed a gradual decrease, consistent with the increased mutational burden and genomic instability associated with disease progression [46]. In contrast, the control samples displayed alignment rates comparable to those of MM, despite no evidence of technical errors, as indicated by the consistent quality checks of both raw and processed reads. The alignment rates for each group are summarized in Table 1.

### 3.3. Mutation Profiles by Type

#### 3.3.1. Functional Mutation Types

The analysis of functional mutation effects, according to Figure 2, revealed that missense mutations were the most prevalent type across all disease stages, suggesting potential impacts on protein structure and function. Other functional categories, such as frameshift mutations, stop-gained mutations, as well as intron and splice site mutations, occur at much lower frequencies throughout all stages. Rare functional classes, including start and stop loss mutations, as well as mutations in untranslated regions (UTRs) and in-frame deletions and insertions, appear only sporadically and were observed in all three stages (full results appear in Appendix B Figure A1).

However, the highest mutation frequency is reported in the SMM stage, contradicting the existing bibliography, which generally reports a higher mutational load in MM [47,48]. Nonetheless, MGUS exhibits lower mutation counts compared to the more advanced stages, and the overall mutation landscape reflects a progressive increase in diversity across disease evolution, in line with previous findings in the literature [6,47].

#### 3.3.2. Structural Mutation Types

The structural classification of mutations is depicted in Figure 3. A clear predominance of single-nucleotide polymorphisms (SNPs) is detected in all disease stages, while insertions (INS) and deletions (DEL) are also present but at much lower frequencies, both patterns aligning with previous studies [6].

Interestingly, the overall mutation burden appears slightly elevated in the SMM stage, compared to the other two stages, as reported previously in functional classifications (Figure 2). Again, this observation diverges from the existing literature, which typically describes a higher mutational load in MM [47,48].

#### 3.3.3. Nucleotide Substitution Patterns

SNPs comprise a variety of base substitutions, which may arise through different mutagenic mechanisms [49]. To characterize mutational patterns across disease progression stages (MGUS, SMM and MM), the substitutions were normalized to reflect their proportion relative to the total number of SNPs observed per stage. According to Figure 4, Cytosine (C) to Thymine (T) transitions were the most frequent type of substitution in MM, with a proportion equal to 0.202, a pattern well documented in multiple cancer types [6,50,51]. This stage-consistent substitution profile supports the hypothesis that such mutational signatures may represent a molecular hallmark of MM pathogenesis. Notably, Guanine (G) to Adenine (A) substitutions were also common across the precancerous stages MGUS and SMM, consistent with the pre-existing literature [50,51].

While some substitution types, were among the most frequently observed across all disease stages, statistical significance (*p*-value < 0.05) was only assigned to those that were disproportionately enriched in a specific stage when compared to others, and are marked with an asterisk in the resulting barplots, as can be seen in Figure 4, which displays normalized substitution proportions per stage with annotated counts.

Across the three conditions, distinct patterns of statistically significant substitution types were observed. In MGUS, both frequent substitutions, like C > T and A > G, and lower-frequency events, namely A > T, T > A, C > G and G > C were significantly enriched, suggesting that hallmark mutations associated with methylation and early mutational processes may emerge at the earliest stages of plasma cell dyscrasias [50]. In contrast, SMM exhibited enrichment primarily in low-frequency nucleotide substitutions, including T > C, G > T, A > T, T > A and G > C, implying a more heterogeneous or transitional mutational profile. Moreover, in MM, significant enrichment was limited to A > G and C > G, potentially reflecting novel mutagenic mechanisms possibly responsible for carcinogenesis. Together, these findings reveal a shift in substitution specificity over the course of disease progression, indicating possible stage-specific mutational mechanisms.

### 3.4. Most Frequently Mutated Genes

To identify genes potentially involved in the progression of the plasma cell dyscrasias examined in this study, the variants that were consistently present across all three disease stages, regardless of the number of patient–control comparisons, were evaluated. The top ten genes identified are shown in Figure 5.

The most recurrent mutations found in all 16 comparisons involved the MHC class I genes, namely *HLA-A*, *HLA-B* and *HLA-C*, which are key components of the immune response and may facilitate immune evasion when mutated [52].

Furthermore, RNF213 protein, is implicated in carcinogenesis, as it has been reported in multiple studies associated with MM and other cancers [53], while, a member of the A-kinase anchor protein family, the *AKAP13* gene, is expressed in a variety of cell types including bone marrow, and is associated with cellular proliferation and oncogenic transformation [54].

Other notable genes include the *MKI67*, a well-established marker of cell proliferation that has been linked to several cancers, including MM and its precursor stage, MGUS [54]. The *SYNE2* is a protein-coding gene, associated with various neoplasms, while another protein-coding gene, the *MAP4*, is involved in microtubule dynamics and its disruption may lead to mitotic abnormalities [54].

Additionally, the *ZMYM5* gene likely contributes to DNA repair processes, and the *BDP1* is implicated in the transcription of small RNAs, which are involved in regulatory and potentially tumorigenic functions [54].

The consistent detection of mutations in these genes across many comparisons suggests a potential role in both the commencement and the progression of MM.

### 3.5. Functional Mutations Emerging During Disease Progression

To highlight the mutations associated with disease progression, only those consistently present across all 16 patient–control comparisons at each transitional phase were selected. This approach ensures that the mutations observed represent stable molecular features of disease advancement rather than sporadic or incidental findings.

As shown in Figure 6, missense variants constitute the predominant functional class during both MGUS to SMM and SMM to MM transitions, which emphasizes their central role in cancer development. Frameshift mutations also appear to contribute to disease progression, further supporting the significance of protein-altering mutations.

In contrast, splice acceptor site and intron variants are detected exclusively during the transition from the precancerous SMM stage to the malignant MM stage, suggesting a possible intensification of transcriptional or epigenetic dysregulation in the later stages of disease evolution [55].

### 3.6. Mutations in Newly Diagnosed Multiple Myeloma Patients

To determine the genes affected among newly diagnosed MM patients, the mutation landscapes of all MM comparisons were analyzed (refer to Appendix A for germline mutations and Appendix A for somatic mutations). The shared germline and somatic genetic alterations, occurring in all 16 patient–control comparisons, are depicted in Figure 7 and Appendix B Figure A2, respectively. The germline analysis revealed only 97 genes consistently altered, whereas in the somatic one, a total of 548 genes were found to be recurrently mutated across all comparisons (refer to Appendix A). Figure 8 depicts the distribution of mutation types among the 548 genes that were consistently identified across all 16 patient–control comparisons.

Germline missense variants were the most prominent mutation class, frequently representing the sole alteration type within individual genes. This prevalence suggests a possible role in inherited susceptibility or modulation of disease risk [54]. Frameshift variants, which introduce significant disruptions to protein-coding sequences, were also observed recurrently. Additional, though less frequent, alterations included stop-gained mutations, disruptive and conservative in-frame deletions, and multi-hit events, which indicate multiple mutations within the same gene in a given comparison. The recurrence and consistency of these germline variants across comparisons suggest potential heritable contributors to MM pathogenesis.

In contrast, the somatic mutations observed in all 16 comparisons showed greater diversity in both type and distribution. Missense variants remained the most common class, highlighting their relevance in tumor evolution. Frameshift mutations also featured prominently, reflecting possible loss-of-function effects. Other detected mutation types included stop and start lost, as well as disruptive in-frame insertions. Notably, multi-hit patterns were more frequent in the somatic context, potentially indicating convergent mutational processes acting on the same gene. Importantly, most genes harbored the same mutation type across all comparisons, suggesting mutation-type exclusivity and possibly selective pressure during disease progression.

To the best of our knowledge, many of the genes identified in both the somatic and germline analysis diverge from previous reported mutation profiles of MM patients [6], suggesting novel or underexplored mechanisms that may be specific to newly diagnosed patients. In certain instances, genes classified in this study as somatic have been previously reported as germline, or they may belong to the same gene family, highlighting possible functional redundancy. Conversely, we also identified molecules consistently reported across studies, reinforcing their established relevance to MM pathogenesis, such as the *KIF* gene and *HLA-A*, *HLA-B* and *HLA-C* antigens [52], which are involved in processes like cell proliferation, immune recognition and DNA repair, further supporting the link between genetic instability and tumorigenesis [52].

Together, these findings underscore a distinction between germline and somatic mutation landscapes. Germline mutations exhibited greater uniformity and potential relevance to inherited predisposition, while somatic mutations reflected broader heterogeneity of mutation types, in line with their dynamic role in clonal evolution and disease advancement [56].

It is also noteworthy that the comparison between germline and somatic mutations revealed both common patterns and substantial differences. While 71 genes were shared between both datasets (e.g., *CPEB4*, *UBE4A*, *ZNF273*, *BIRC6*, *SETBP1*), 26 genes were exclusively mutated in the germline context, and 477 genes were unique to the somatic dataset.

Moreover, some of the genes highlighted in this study diverge slightly from previously published mutation profiles [6]. These discrepancies may reflect multiple factors, including differences in patient cohorts, disease subtypes, population-specific genomic backgrounds, or the specific context of newly diagnosed MM patients, as analyzed in this dataset. The early-stage mutational landscape may differ from that of relapsed or refractory cases, potentially capturing distinct biological pathways active at disease onset [57].

## 4. Discussion

In this study, we analyzed single-cell RNA-sequencing (scRNA-seq) data from patients diagnosed with Monoclonal Gammopathy of Undetermined Significance (MGUS) and Smoldering Multiple Myeloma (SMM), as well as newly diagnosed patients with Multiple Myeloma (MM), to investigate the mutational landscape of plasma cell dyscrasias, with a focus on the later disease, by utilizing a comprehensive bioinformatics pipeline tailored for scRNA-seq data. High-quality preprocessing, followed by variant calling and the application of strict filtering thresholds, enabled the identification of genetic alterations across disease stages. Our findings confirmed that missense variants were the most prevalent functional type of mutations, both in early precursor stages and in MM, with additional contributions from intron and splice-site mutations emerging during the tumorigenic stage. Notably, somatic variants found in MM datasets exhibited greater functional mutation-type diversity, while germline variants were more uniform, often dominated by missense mutations alone. Importantly, mutations present across all 16 patient–control comparisons likely represent stable molecular hallmarks of disease and point to underlying mechanisms of pathogenesis. Although germline mutations, which are inherited, play a role in certain forms of cancer, the majority of cancer cases are attributed to somatic mutations that accumulate over time due to endogenous processes or environmental factors [56]. These findings support the theory of a synergistic effect between predisposing and acquired genetic factors in the development of MM and the oncogenesis that characterizes this stage. All in all, the complexity of each patient’s molecular profile demonstrates that different combinations of gene mutations drive the progression of plasma cell dyscrasias and cancers in general.

In contrast to prior reports describing a higher mutational load in MM, our analysis revealed an unexpectedly elevated mutation burden in the intermediate SMM stage [47,48]. While previous studies often rely on bulk DNA-seq or exome-level profiling, our use of scRNA-seq data captured different molecular signals, including low-frequency subclonal events [58]. Additionally, the identification of 548 somatic and 97 germline genes mutated in all MM comparisons reflects a robust core of consistently altered genes, while the 71 overlapping genes between germline and somatic datasets further underscore a degree of mutation-type specificity. Furthermore, the large amount of novel identified genes in newly diagnosed MM patients suggests underexplored mechanisms that may be specific to newly diagnosed patients and may result in the progression of the disease [6]. Even so, the identified key mutated genes, such as the HLA genes and members of the KIF, EP400 and KDM families, may not only serve as potential biomarkers but also provide insights into the molecular mechanisms driving disease progression and highlight targets for future therapeutic treatments.

Our results reveal mechanistic insights that expand upon prior studies. The emergence of splice-site and intron variants specifically during the SMM to MM transition suggests that transcriptional dysregulation becomes increasingly pronounced in later disease stages [55]. Unlike the more heterogeneous mutation profiles reported in relapsed MM cases, the relative uniformity seen here may reflect a mutational foundation established early in disease. Moreover, the analytical pipeline utilized in this study is designed to be accessible and may be readily adopted without extensive bioinformatics expertise, while remaining fully functional for retrieving and interpreting variant information.

Several methodological limitations should be considered when interpreting our findings. A major constraint was the lack of detailed clinical metadata, particularly for control samples [59], and more generally for all samples utilized in this study, which limited our ability to account for confounding factors, such as other medical conditions or medication, which may have influenced the variant calling process. In addition, the relatively small number of samples per stage also limits the extent to which the findings can be extrapolated to the wider population. Equally important is the fact that all MM cases analyzed were from newly diagnosed patients, who may exhibit lower mutational burden compared to relapsed cases. Moreover, the novel mutations identified in our study vary from previously published mutation profiles, which may reflect multiple factors, including the absence of clinical metadata and the omission of a deduplication step to prevent the exclusion of biologically important repetitive regions. These regions are difficult to differentiate from potential technical artifacts in scRNA-seq data and, thus, cannot be entirely ruled out when interpreting our results. While the pipeline functioned effectively for mutation identification in this dataset, it may not be directly transferable to other sequencing platforms or methods. Nevertheless, rigorous quality control and filtering steps were applied throughout the workflow to ensure the reliability and validity of the results.

While the results of this study are promising, further work is needed to build upon these. Although these findings underscore critical points in disease evolution that could be leveraged for early treatment or disruption of malignant progression, future studies should involve larger and more diverse patient cohorts, including relapsed cases, to explore resistance mechanisms and validate key alterations. Furthermore, additional analyses could enhance the functional and biological interpretation of the identified variants in our dataset, including the assessment of their biological impact, using oncogenic signaling pathways or the Gene Set Enrichment Analysis to identify enriched biological pathways or functional categories. The analysis of copy number variation (CNV) could further uncover important genomic insights that may prove valuable to premature MM detection. In addition, future work could expand on our current variant annotation approach by incorporating alternative tools, such as ANNOVAR 2019 (https://annovar.openbioinformatics.org/en/latest/, accessed on 8 May 2025), alongside SnpEff, enabling a comparative evaluation of their outputs and potentially refining variant interpretation in the context of MM. Finally, mutational signature analyses and the identification of potential driver genes would strengthen both the clinical and biological significance of the findings.

## 5. Conclusions

The single-cell RNA-sequencing-based mutation analysis revealed meaningful molecular alterations in early stages of Multiple Myeloma, offering insights into early disease mechanisms and the transition from precursor stages. By distinguishing between germline and somatic mutations and identifying stage-specific events, we provide insight into the genetic basis of disease progression. These findings contribute to a deeper understanding of Multiple Myeloma evolution and may inform future strategies for early detection and targeted intervention.

## Figures and Tables

**Figure 1 diagnostics-15-02130-f001:**
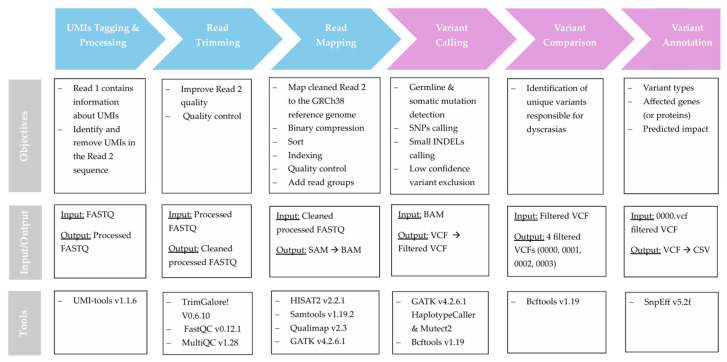
Workflow pipeline for preprocessing (blue arrows) and variant detection (pink arrows), indicating the purpose of each step, the tools used, and the corresponding files. Black arrows within the boxes represent the conversion of input files into corresponding output files at each stage.

**Figure 2 diagnostics-15-02130-f002:**
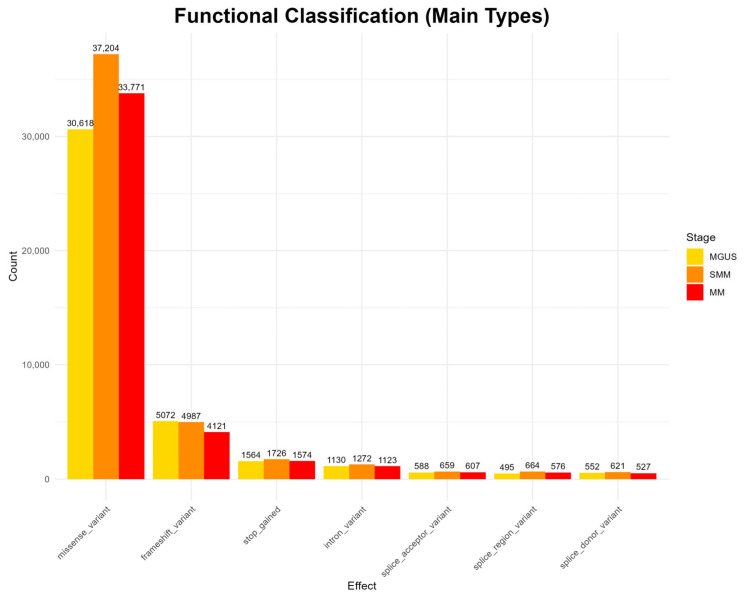
Functional classification of variants detected at each disease stage. Only the main functional mutation types are displayed; the full classification is provided in Appendix B, Figure A1.

**Figure 3 diagnostics-15-02130-f003:**
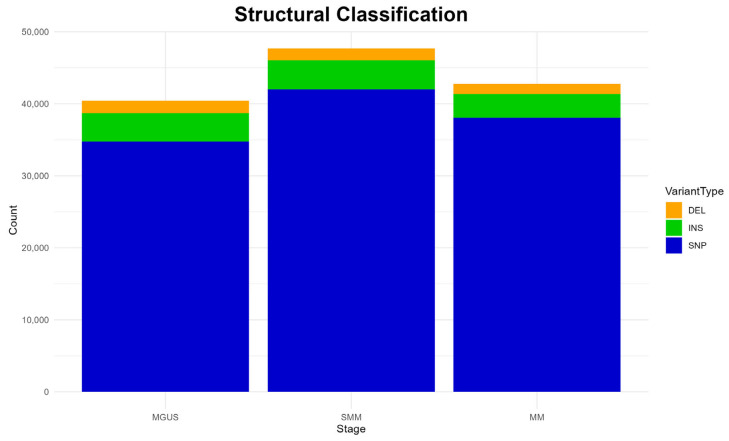
Structural classification of mutations detected at each disease stage.

**Figure 4 diagnostics-15-02130-f004:**
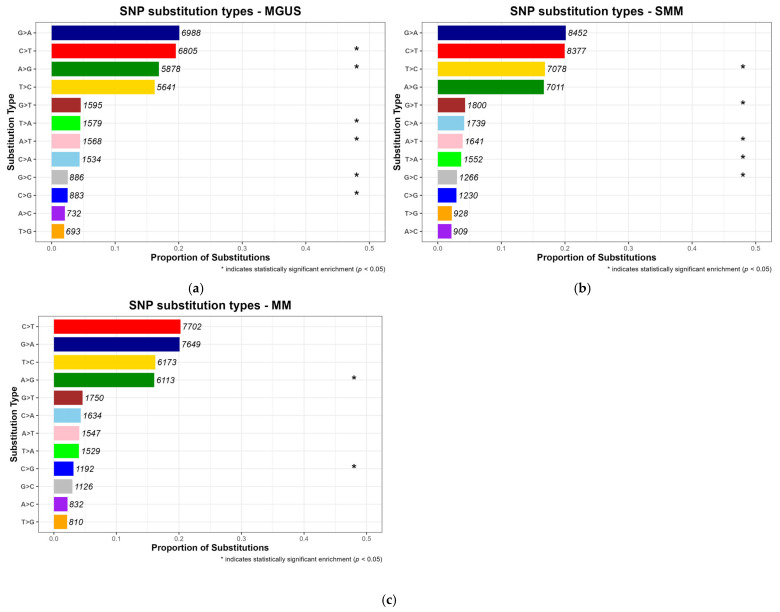
Normalized proportions of SNP substitution types across disease stages, (**a**) MGUS, (**b**) SMM and (**c**) MM. Each panel displays the relative frequency of single-nucleotide substitution types normalized to the total number of SNPs in that stage. Numeric labels represent raw substitution counts. Substitution types marked with an asterisk (*) were found to be significantly enriched in that stage (*p* < 0.05, Fisher’s exact test), indicating that their frequency was statistically higher compared to their frequency in the other two stages.

**Figure 5 diagnostics-15-02130-f005:**
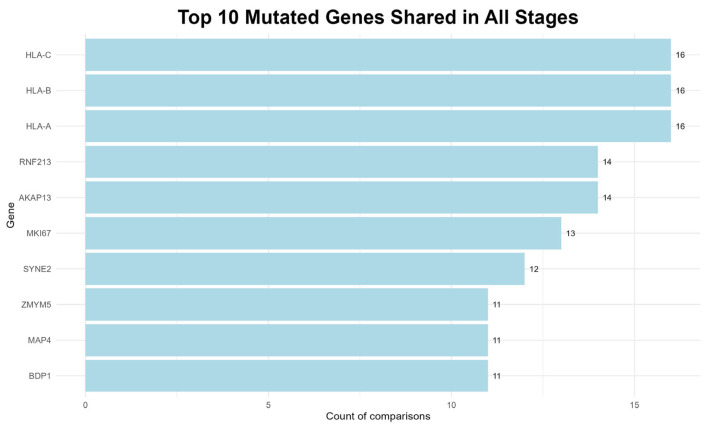
The top ten most frequently mutated genes, common to all disease stages.

**Figure 6 diagnostics-15-02130-f006:**
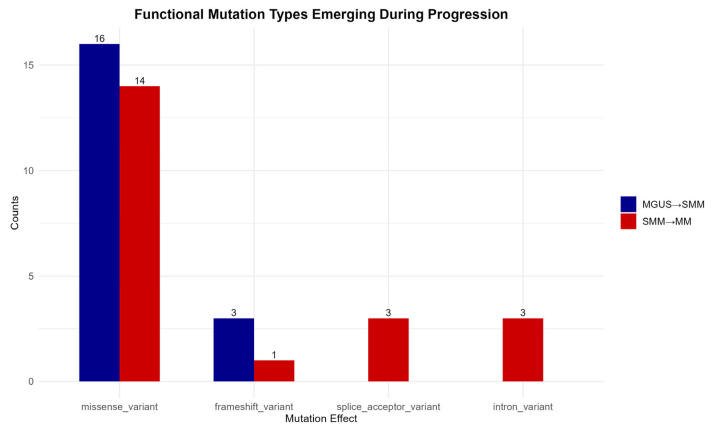
Functional mutation types consistently present during stage-to-stage transitions in disease progression, with arrows indicating stage transitions.

**Figure 7 diagnostics-15-02130-f007:**
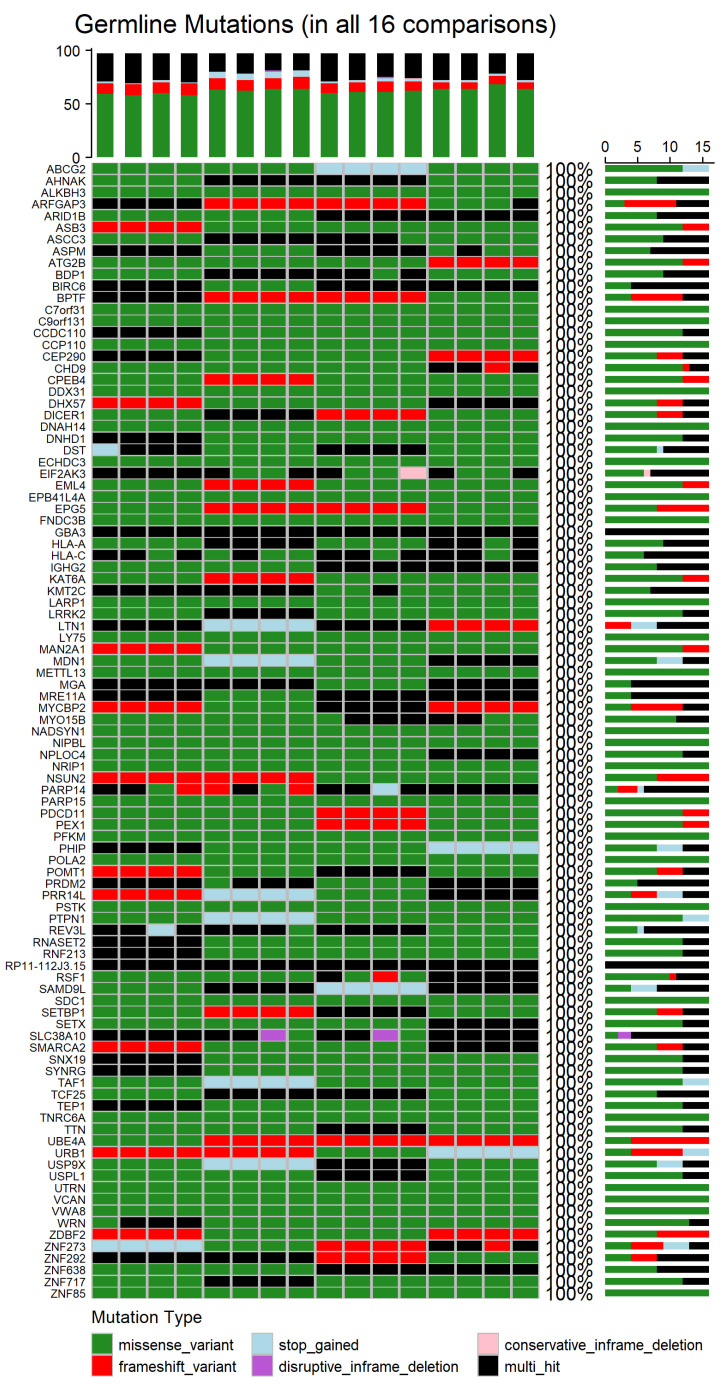
Oncoprint of germline mutations shared across all Multiple Myeloma patient–control comparisons. Mutation types are color-coded; each box from left to right represents each comparison. The bar plot above indicates the number of mutations per comparison, while the percentage on the right shows the frequency of each gene mutation across the 16 comparisons.

**Figure 8 diagnostics-15-02130-f008:**
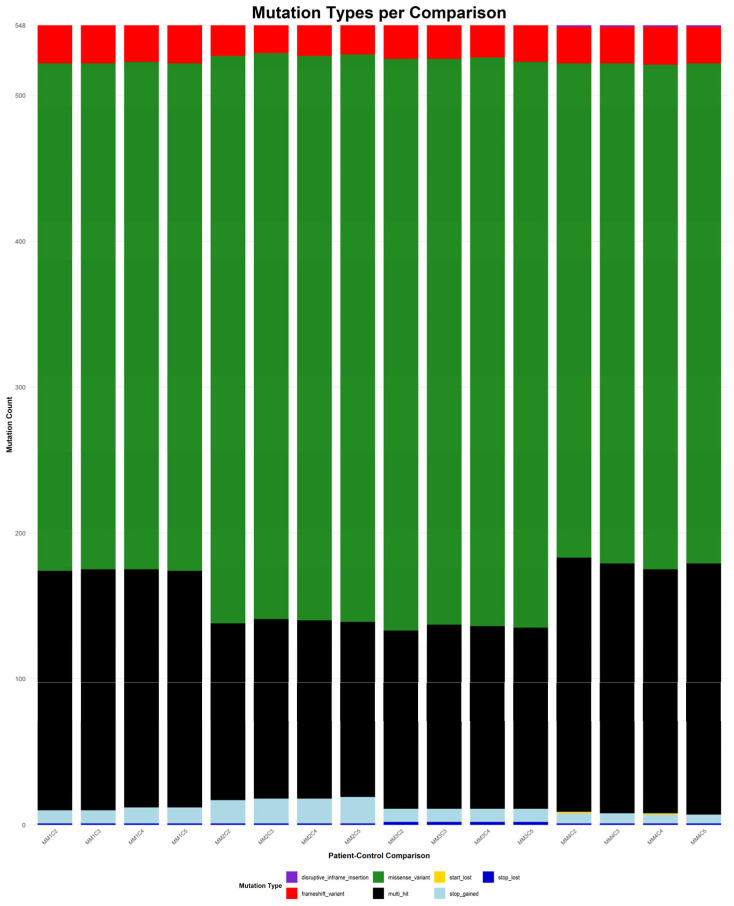
Barplot showing the distribution of mutation types among the 548 somatic genes that were consistently identified across all 16 MM patient–control comparisons. Each bar represents one comparison, and the segments are color-coded according to mutation type. The height of each bar reflects the number of shared mutated genes per comparison, while the color composition illustrates the relative contribution of each mutation category.

**Table 1 diagnostics-15-02130-t001:** Overall alignment rates (percentages) for each individual sample, grouped by sample category, as well as the mean alignment rate, are reported for each group. Values represent the percentage of reads successfully aligned to the GRCh38 reference genome.

Sample	MGUS (%)	SMM (%)	MM (%)	Controls (%)
1	92.51	84.71	90.27	91.42
2	91.84	94.25	89.72	92.72
3	89.05	83.43	83.63	85.01
4	87.65	88.24	88.39	83.11
Mean	90.26	88.24	88.00	88.07

## Data Availability

The datasets analyzed in this study are available in GEO under the following accession numbers: (a) MGUS patients: GSM8369868, GSM8369869, GSM8369870, GSM8369871; (b) SMM patients: GSM8369874, GSM8369875, GSM8369876, GSM8369877; (c) newly diagnosed MM patients: GSM8369878, GSM8369879, GSM8369880, GSM8369881; (d) controls: GSM8369864, GSM8369865, GSM8369866, GSM8369867. The code implemented in this study is available upon request.

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
