# Peer review of "Genomic Insights into Tumorigenesis in Newly Diagnosed Multiple Myeloma"

_diagnostics, 2025, doi:10.3390/diagnostics15172130_

Round 1
Reviewer 1 Report
Comments and Suggestions for Authors
1、Please clarify the prevalence of MM specifically within the European population. Including this epidemiological context would enhance the relevance of your findings and provide a better understanding of the disease burden in the studied cohort.
2、It would be helpful to define and discuss the known risk factors associated with the progression of MM, particularly in the context of your cohort. How do these align with your findings?
3、Could you elaborate on the rationale for choosing both Trimmomatic and Cutadapt for your data preprocessing? What specific parameters were used, and on what basis were they selected?
4、Are you following any established guidelines or best practices for filtering annotated variants? Please clarify the criteria used (e.g., read depth, allele frequency thresholds, population databases).
5、I suggest performing a GSEA to explore whether specific biological pathways or functional categories are enriched in your dataset. This would add functional relevance to your variant analysis.
6、You have used SnpEff for variant annotation. Is there a specific reason for preferring SnpEff over other tools like ANNOVAR? Given the nature of your data, ANNOVAR may offer more comprehensive annotation options and database support.
7、Consider utilizing your annotated variants to estimate alterations in oncogenic signaling pathways (e.g., using MSigDB or KEGG pathways). This would help interpret the biological impact of the variants identified.
8、The Oncoprint shows a 100% mutation frequency across all 16 genes, which is highly unusual. Please clarify how this was determined and whether these frequencies reflect true biological signals or potential technical artifacts.
9、Please consider including Copy Number Variation (CNV) and mutational signature analyses. These are critical components of comprehensive cancer genome profiling and could uncover additional insights into MM pathogenesis.
10、It would be valuable to identify potential driver genes from your dataset using tools like OncodriveCLUST, MutSigCV, or dNdScv. This would strengthen the clinical and biological significance of your findings.
Reviewer 2 Report
Comments and Suggestions for Authors
More information need to be provided in order to characterize evaluated material. Both MGUS, SMM, and MM are heterogenous entities that are initially characterized by type of monoclonal protein produced and by significant genetic changes such hyperdiploidy or various chromosome translocations and deletions.
Furthermore, the most affected are HLA genes: would this mean the appearance of novel HLA specificities?
